# SOLVING AND LEARNING NON-MARKOVIAN STOCHASTIC CONTROL PROBLEMS IN CONTINUOUS-TIME WITH NEURAL RDES

## ABSTRACT

We propose a novel framework for solving continuous-time, non-Markovian stochastic optimal problems with the use of neural rough differential equations (Neural RDEs). By parameterising the control process as the solution of a Neural RDE driven by the state process, we show that the control-state joint dynamics are governed by an uncontrolled RDE with structured vector fields, allowing for efficient trajectories simulation, Monte-Carlo estimation of the value function and backpropagation. To deal with input paths of infinite 1-variation, we refine the universal approximation result in Kidger et al. (2020) to a probabilistic density result for Neural RDEs driven by random rough paths. Experiments on various non-Markovian problems indicate how the proposed framework is time-resolution-invariant and capable of learning optimal solutions with higher accuracy than traditional RNN-based approaches. Finally, we discuss possible extensions of this framework to the setting of non-Markovian continuous-time reinforcement learning and provide promising empirical evidence in this direction.

## 1 INTRODUCTION

The field of stochastic control is concerned with problems where an agent interacts over time with some random environment through the action of a *control*. In this setting, the agent seeks to select the control such that some objective depending on the trajectory of the system under their control and the choice of the control itself is optimised; commonly, as the system is stochastic, such an objective takes the form of an expectation of some pathwise cost or reward. The study of this class of problems is intimately related to reinforcement learning (RL) and has been successfully applied to many fields of modern sciences, including biology Cucker & Smale (2007), economics Kamien & Schwartz (2012), engineering Grundel et al. (2007), finance Pham (2009), and more recently, epidemics control Hubert et al. (2022).

Stochastic control is nowadays regarded as a well-established field of mathematics. Two main approaches govern the analysis: the stochastic maximum principle and the dynamic programming approach, see Yong & Zhou (1999); Pham (2009). In either case, an agent is interested in characterising a set of optimal strategies, the dynamics of the system under such strategies, and the optimal value of the corresponding reward functional. The two main sources of complexity for tackling these problems are: 1) the continuous-time nature of the underlying stochastic dynamics, and 2) the presence of memory yielding a non-negligible impact of the system's history on its future evolution.

On the one hand, compared to their discrete counterparts, continuous-time stochastic control problems have received an increasing amount of attention in recent years, partly because the underlying physical processes themselves often develop in continuous time, partly because of their characterisation via partial differential equations (PDEs) or backward stochastic differential equations (BSDEs).

On the other hand, non-Markovian stochastic control problems, where the evolution of the system depends on its history and not only on its current state, often provide a more faithful class of models to describe real-world phenomena than their Markov counterparts, where the (infinitesimal) displacement of the state dynamics depend only on the current state.

Typical examples of settings where non-Markovian stochastic control problems in continuous-time arise include rough volatility models Gatheral et al. (2018) from quantitative finance in which the non-Markovianity stems from having a fractional Brownian motion as the driving noise. Another common source of non-Markovian problems are delayed control problems, where memory is incorporated into the system by assuming path-dependence of the vector fields governing the dynamics (see Sec. 3 for a precise statement). These are ubiquitous in economics, for example in the study of growth models with delayed production or pension funds models, Kydland & Prescott (1982); Salvatore (2011), in marketing for models of optimal advertising with "distributed lag" effects Gozzi et al. (2009), and in finance for portfolio selection under the market with memory and delayed responses Øksendal et al. (2011). See also Kolmanovskiı & Shaıkhet (1996) for modelling systems with after-effect in mechanics, engineering, biology, and medicine.

Despite recent theoretical advances in simplified settings, non-Markovian stochastic control problems in continuous-time are often not analytically tractable, a fact that undeniably motivates the need for developing efficient numerical schemes to solve them. Additionally, such methods could provide a fruitful basis for (non-Markovian) reinforcement learning in continuous time, studied in the Markovian case recently by Jia & Zhou (2021); Wang et al. (2020).

**Contributions**   Using the modern tool set offered by neural rough differential equations (Neural RDEs) Morrill et al. (2021) — a continuous-time analogue to recurrent neural networks (RNNs) — we propose a novel framework which, to the best of our knowledge, is the first numerical approach allowing to solve non-Markovian stochastic control problems in continuous-time. More precisely, we parameterise the control process as the solution of a Neural RDE driven by the state process, and show that the control-state joint dynamics are governed by an uncontrolled RDE with vector fields parameterised by neural networks. We demonstrate how this formulation allows for trajectories sampling, Monte-Carlo estimation of the reward functional and backpropagation. To deal with sample paths of infinite 1-variation, which is necessary in stochastic control, we also extend the universal approximation result in Kidger et al. (2020) to a probabilistic density result for Neural RDEs driven by random rough paths. The interpretation is that we are able to approximate continuous feed-back controls arbitrarily well in probability. Through various experiments, we demonstrate how the proposed framework is time-resolution-invariant and capable of learning optimal solutions with higher accuracy than traditional RNN-based approaches. Finally, we discuss possible extensions to the setting of non-Markovian reinforcement learning (RL) in continuous-time and provide promising empirical evidence in this direction.

The rest of the paper is organised as follows: in Sec. 2 we discuss some related work, in Sec. 3 we present our numerical scheme and the universality result, in Sec. 4 we study the extension to non-Markovian RL in continuous-time, and in Sec. 5 we present our numerical results.

## 2   RELATED WORK

Over the last decade, a large volume of research has been conducted to solve Markovian stochastic control problems numerically using neural networks, either by directly parameterising the control and then sampling from the state process, such as done by Han et al. (2016), or by solving the PDEs or BSDEs associated with the problem; see Germain et al. (2021) for a recent survey about neural networks-based algorithms for stochastic control and PDEs. We also mention two examples from the growing literature. The Deep BSDE model from Han et al. (2017), where the authors propose an algorithm to solve parabolic PDEs and BSDEs in high dimension and think of the gradient of the solution as the policy function, approximated with a neural network. The Deep Galerkin model Sirignano & Spiliopoulos (2018) is a mesh-free algorithm to solve PDEs associated with the value function of control problems; the authors approximate the solution with a deep neural network which is trained to satisfy the PDE differential operator, initial condition, and boundary conditions.

Recently, signatures methods Lyons (2014); Kidger et al. (2019) have been employed for solving both Markovian and non-Markovian control problems in simplified settings Kalsi et al. (2020); Cartea et al. (2022). This approach does not rely on a model underpinning the dynamics of the unaffected processes and has shown excellent results when solving a number of algorithmic trading problems. However, this method has two main drawbacks: (i) it suffers from the curse of dimensionality — this happens when one wishes to compute signatures of a high-dimensional (more than five)

process to make online decisions, and (ii) it requires that the flow of information observed by the controller is unaffected by the control and everything else the controller observes can be explicitly constructed from such information and the policy. We also point out the theoretical contribution by Diehl et al. (2017) studying control problems where the driving noise is a random rough path.

The approach of directly parameterising the control and training by sampling trajectories from the system was recently studied in the setting of delay-type non-Markovian stochastic control by Han & Hu (2021). Specifically, the control is taken to be a Long Short-Term Memory (LSTM) recurrent neural network with the discrete simulated values of the state process as input, so as to capture the path-dependence of the problem. The method is shown to outperform a baseline parameterisation using a fully-connected feed-forward network taking as input a segment of the history of the sample path, and demonstrated to have theoretical advantages in handling non-Markovian problems.

Neural RDEs, as popularised by Kidger et al. (2020); Morrill et al. (2021) provide an elegant way of modelling temporal dynamics by parameterising the vector fields of some classes of differential equations by neural networks. The input to such models is a multivariate time series interpolated into a continuous path $X$. Depending on the level of (ir)regularity of $X$, the corresponding equation can be solved in different ways. In (Kidger et al., 2020), $X$ is assumed differentiable almost everywhere, and the equation becomes an ordinary differential equation (ODE) that can be evaluated numerically via a call to an ODE solver of choice. More generally, if $X$ is of bounded variation, then the Neural RDE can be solved using classical Riemann–Stieltjes or Young integration (Young, 1905).

Of particular interest in the field of stochastic control is the setting where the driving noise is Brownian motion , and the resulting dynamical systems are typically referred to as *stochastic differential equations* (SDEs). Because sample paths from Brownian motion are not of bounded variation, the integral cannot be interpreted in the classical sense, but rather using the framework of stochastic integration (Itô, Stratonovich, etc.). The corresponding "neural" version of such models has been the object of several studies Liu et al. (2019); Li et al. (2020); Kidger et al. (2021b;a), in particular in the context of generative modelling for time series. Rough integration Lyons (1998) is arguably the most general type of integration theory accommodating driving signals $X$ of arbitrary roughness, and in particular non-Markovian processes such as fractional Brownian motion. In this paper, we position ourselves in this general setting. In the appendix, we provide a minimal summary of the basic notions of this theory underpinning the content of this paper.

## 3 METHOD

### 3.1 STOCHASTIC CONTROL PROBLEMS WITH PATH-DEPENDENT COEFFICIENTS

Let us introduce the non-Markovian control problems over closed-loop controls. We fix $d, d_a, d_W \in \mathbb{N}$, a real number $T > 0$ and $\mathcal{C}^d := \mathcal{C}([0, T]; \mathbb{R}^d)$, the space of continuous paths from $[0, T]$ to $\mathbb{R}^d$ endowed with the sup norm. Let $(\Omega, \mathcal{F}, \mathbb{P})$ be a probability space supporting a $d_W$-dimensional Brownian motion $W = (W_t)_{t \in [0,T]}$, and $\mathbb{F}$ be the natural filtration of $W$ augmented with the $\mathbb{P}$-null sets. Let $\mathcal{H}^2(\mathbb{R}^{d_a})$ be the space of all square integrable $\mathbb{F}$-progressively measurable processes, and for each $\alpha \in \mathcal{H}^2(\mathbb{R}^{d_a})$, consider the following controlled state dynamics:

$$\mathrm{d}X_t = \mu(t, X_{\cdot \wedge t}, \alpha_t)\mathrm{d}t + \sigma(t, X_{\cdot \wedge t}, \alpha_t)\mathrm{d}W_t, \ t \in [0, T]; \quad X_0 = x_0, \tag{1}$$

where $X_{\cdot \wedge t} = \{X_s\}_{s \in [0,t]}$, $(\mu, \sigma) : [0, T] \times \mathcal{C}^d \times \mathbb{R}^{d_a} \longrightarrow \mathbb{R}^d \times \mathbb{R}^{d \times d_W}$ are non-anticipative and sufficiently regular mappings so that equation (1) admits a unique solution $X$ in $\mathcal{H}^2(\mathbb{R}^d)$.[1] We denote by $\mathcal{A}$ the set of admissible controls containing all $\alpha \in \mathcal{H}^2(\mathbb{R}^{d_a})$ that are adapted to the filtration generated by $X$. Such controls are often referred to as closed-loop, or feedback, controls.

The agent's goal is to minimise the following objective functional

$$J(x_0, \alpha) = \mathbb{E}\left[\int_0^T f(t, X_{\cdot \wedge t}, \alpha_t)\mathrm{d}t + g(X_{\cdot \wedge T})\right] \tag{2}$$

over all closed-loop controls $\alpha \in \mathcal{A}$, where $f : [0, T] \times \mathcal{C}^d \times \mathbb{R}^{d_a} \to \mathbb{R}$ and $g : \mathcal{C}^d \to \mathbb{R}$ are given measurable functions.

---

[1] This is the case if, for instance, $\mathcal{C}^d \times A \ni (x, a) \longmapsto \varphi(t, x, a)$ has linear growth and is Lipschitz continuous uniformly in $t$ for $\varphi = \mu, \sigma$, see Protter (2005).

### 3.2 A MODEL-BASED APPROACH

Here, we are going to parameterise the control process $\alpha$ in equation (1) as the solution of a Neural RDE driven by the state process $X$. Let $\ell_\theta : \mathbb{R}^{d_a} \to \mathbb{R}^{d_h}, h_\theta : \mathbb{R}^{d_h} \to \mathbb{R}^{d_h \times d}, A_\theta \in \mathbb{R}^{d \times d_h}$ be (Lipschitz) neural networks. Collectively, they are parameterised by $\theta$. The dimension $d_h > 0$ is a hyperparameter describing the size of the hidden state.

We parameterise controls $\alpha^\theta \in \mathcal{A}$ as solutions to Neural RDEs driven by $X$,

$$Y_0 = \ell_\theta(x_0), \quad \mathrm{d}Y_t = h_\theta(Y_t)\mathrm{d}X_t, \quad \alpha_t^\theta = A_\theta Y_t. \tag{3}$$

With this choice of parameterisation, the dynamics of the joint process $(X, Y)$ are governed by the following *uncontrolled* RDE with structured vector fields

$$\mathrm{d}\begin{pmatrix} X \\ Y \end{pmatrix}_t = \mu\left(t, X_{\cdot \wedge t}, A_\theta Y_t\right)\begin{pmatrix} 1 \\ h_\theta(Y_t) \end{pmatrix}\mathrm{d}t + \sigma\left(t, X_{\cdot \wedge t}, A_\theta Y_t\right)\begin{pmatrix} I_d & 0 \\ 0 & h_\theta(Y_t) \end{pmatrix}\mathrm{d}\begin{pmatrix} W \\ W \end{pmatrix}_t \tag{4}$$

Thus, the infinite dimensional minimisation over admissible controls of the reward functional $J$ in equation (2) can be replaced with the finite-dimensional minimisation over the parameters $\theta$ of the following objective functional

$$J(x_0, \alpha^\theta) = \mathbb{E}\left[\int_0^T f(t, X_{\cdot \wedge t}, A_\theta Y_t)\mathrm{d}t + g(X_{\cdot \wedge T})\right], \tag{5}$$

Here, we perform this minimisation by first solving numerically the uncontrolled Neural RDE (4) using a classical Euler-Maruyama scheme[2]; we then use the obtained sample trajectories to compute a Monte-Carlo estimate of the objective functional in (5), where the integral is approximated using classical quadrature; finally we compute gradients of the estimated objective functional with respect to model parameters $\theta$ and optimise by (stochastic) gradient descent.

Contrary to the approach taken by Han & Hu (2021) using an LSTM-parameterisation of the control, our formulation does not rely on any specific discretisation or choice of numerical method. A key feature of Neural RDEs is their robustness to irregular sampling of the data, essentially due to operating continuously in time. The sampled data enters the model only through the construction of the interpolated path, after which the RDE can be solved numerically on any desired grid using adaptive schemes that changes the step size to appropriately resolve the variations in the path. Therefore, because our scheme can be formulated completely in continuous-time and independently of whichever way one chooses to estimate $J(\alpha)$, it is naturally time-resolution invariant, so that even if trained on a coarser resolution it can be directly evaluated on a finer resolution without retraining.

### 3.3 UNIVERSALITY

Unless $\sigma \equiv 0$ in equation (4), $X$ will not be of bounded variation and its support is generally not a compact set. Therefore, the universal approximation result for Neural RDEs given by (Kidger et al., 2020, Theorem B.7.) does not apply. In the next theorem, formally stated and proved in the Appendix (see Theorem A.1), we reformulate this universality result as a probabilistic density result for Neural RDEs driven by random paths of arbitrary (ir)regularity (or rough paths).

**Theorem 3.1** (**Informal**). *The action of a linear map on the terminal value of a Neural RDE is a universal approximator, in probability, from random rough paths to $\mathbb{R}$.*

The interpretation of this result in the present setting is that we are able to approximate any continuous feed-back control arbitrarily well, in probability, using the proposed Neural RDE method.

### 3.4 EXTENSION TO CONTROL PROBLEMS WITH NON-MARKOVIAN NOISE

Up until now we have considered stochastic control problems where the non-Markovianity stems from some path-dependence of the coefficients on the history of the system. Indeed, the heuristic meaning of equation (1) is that the infinitesimal increment $X_{t+\mathrm{d}t} - X_t$ is normally distributed with

---

[2]For convergence guarantees of Euler-Maruyama schemes applied to SDEs with path-dependent vector fields we refer the reader to Mao (2003). Other choices of solvers are possible.

mean $\mu(t, X_{\cdot \wedge t}, \alpha_t)dt$ and variance $\sigma(t, X_{\cdot \wedge t}, \alpha_t)^2 dt$, and independent of the past $\mathcal{F}_t$: the solution is not Markovian because knowing the value of $X_t$ does not contain all the information necessary to evaluate the path-dependent coefficients $\mu$ and $\sigma$, which are needed to compute mean and variance of the increment. There is a second, very different way in which the process $X$ can fail to satisfy the Markov property: consider equation (1) in the case in which $W$ is a process with correlated increments such as fractional Brownian motion $W^H$. Now Markovianity does not hold because the increment $X_{t+dt} - X_t$ is not independent of the past, even if $\mu$ and $\sigma$ are state-dependent: this is because the noise increment $dW^H$ is correlated with its own history, even conditionally on the present. However, the approach in Sec. 3.2 can be naturally extended to this setting, as the algorithm directly parameterizes the feedback controls and hence is invariant up to different driving noises.

## 4 TOWARDS NON-MARKOVIAN REINFORCEMENT LEARNING IN CONTINUOUS-TIME

In this section we describe a heuristic extension of the framework developed in Sec. 3 to the setting of non-Markovian RL in continuous-time. In what follows, we will assume that the state dynamics of equation (1) are not known to the user, but instead we will assume that the user has the ability to interact with a random environment and generate a collection of state-action trajectories to learn from. We begin by reminding the classical Markovian RL setup in discrete-time.

**Markovian RL in discrete-time** In RL, it is typical to consider a Markov Decision Process (MDP) where for any two states $x, x' \in \mathbb{R}^d$ and action $a \in \mathbb{R}^{d_\alpha}$, the quantity $\mathbb{P}(x'|a, x)$ denotes the probability that the system transitions from state $x$ to $x'$ given action $a$. The agent's actions are represented by a parametric (deterministic) policy $p_\theta : \mathbb{R}^d \to \mathbb{R}^{d_\alpha}$ that, upon interaction with the Markov environment, generates a collection of state-action discrete sequences

$$\left\{ \begin{matrix} x^i_{t_0}, ..., x^i_{t_n} \\ a^i_{t_0}, ..., a^i_{t_n} \end{matrix} \right\}^N_{i=1} \quad \text{such that} \quad \begin{cases} x^i_{t_{k+1}} \sim \mathbb{P}(x'|x = x^i_{t_k}, a = a^i_{t_k}) \\ a^i_{t_k} = p_\theta(x^i_{t_k}) \end{cases}$$

Finally, the agent aims at maximising an expected final reward $\mathbb{E}\left[\sum_{k=0}^n R(x_{t_k}, p_\theta(x_{t_k}))\right]$. In deep RL, $p_\theta$ is commonly parameterised by a neural network and the optimisation is carried out by (stochastic) gradient descent. The gradient $\nabla_\theta J(\pi_{\theta_t})$ is usually not directly computable, and policy gradients methods allow to approximate it (PPO, TRPO etc.) Schulman et al. (2015; 2017).

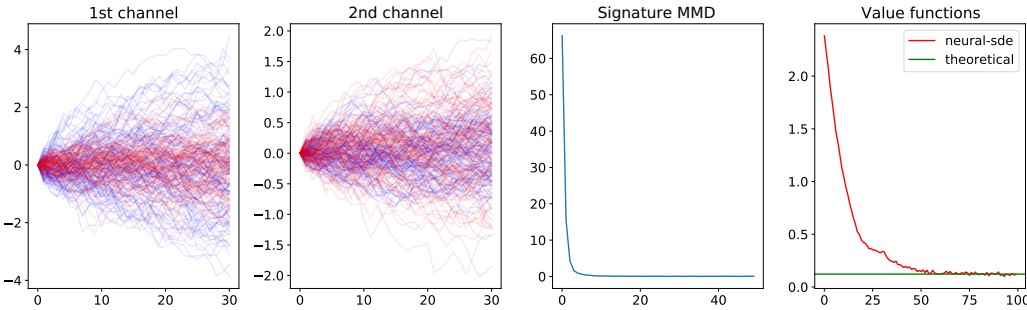

Figure 1: The first and second plot from the left illustrate the dynamics of the first two channels of the true process $X$ (in blue) and of the learnt process $X^\phi$ (in red). The third plot from the left indicates the convergence of the signature MMD to a value close to 0 during the first part of the two-step optimisation described in this section, indicating that the learnt dynamics of the process $X^\phi$ are, statistically speaking, close to the dynamics of the true process $X$ as per equation (8). The last plot on the right shows the convergence of the estimate of the value function $J(\alpha^\theta)$ obtained by subsequently training the parameters $\theta$ of the control process $\alpha^\theta$ in equation (7).

**Non-Markovian RL in continuous-time** In a non-Markovian, continuous-time setting, we assume that the dynamics for the state variable $X$ are governed by an unknown controlled diffusion

process with path-dependent vector fields [3]. The (deterministic) policy is now a function of the form $C([0,T], \mathbb{R}^d) \to \mathbb{R}^{d_\alpha}$, taking as input a continuous trajectory of states, and producing as output a configuration of actions. Thanks to the results established in Sec. 3 we know that Neural RDEs form a dense class of approximators for functions on rough paths, thus it is natural to parameterise the policy as a Neural RDE. Upon interaction with the non-Markovian environment, we obtain a collection of state-action continuous sample trajectories

$$\left\{ \begin{array}{l} X^i : [0,T] \to \mathbb{R}^d \\ \alpha^i : [0,T] \to \mathbb{R}^{d_\alpha} \end{array} \right\}_{i=1}^N \text{ s.t. } \left\{ \begin{array}{l} X^i \sim \mathrm{d}X_t = \mu(t, X_{\cdot \wedge t}, \alpha_t)\mathrm{d}t + \sigma(t, X_{\cdot \wedge t}, \alpha_t)\mathrm{d}W_t \\ \alpha^i \sim A_\theta Y_t, \ Y_0 = \ell_\theta(X_0), \ Y_t = Y_0 + \int_0^t h_\theta(Y_s)dX_s \end{array} \right. \quad (6)$$

As in the stochastic control setting of Section 3, the agent aims at maximising and expected final reward $J(\alpha^\theta)$ as given in equation (5). This time however, because the controlled dynamics of $X$ are unknown, it is not possible to directly compute gradients of $J(\alpha^\theta)$ with respect to $\theta$ and optimise the reward functional $J$ by backpropagation.

To overcome this issue, we parameterise the unknown drift $\mu$ and diffusion $\sigma$ in equation (6) by neural networks $\mu_\phi$ and $\sigma_\phi$ and train a second Neural RDE as a generative model in order to learn the unknown dynamics of the random environment. In this way, equation (4) becomes

$$\mathrm{d}\begin{pmatrix} X \\ Y \end{pmatrix}_t = \mu_\phi\left(t, X_{\cdot \wedge t}, A_\theta Y_t\right)\begin{pmatrix} 1 \\ h_\theta(Y_t) \end{pmatrix}\mathrm{d}t + \sigma_\phi\left(t, X_{\cdot \wedge t}, A_\theta Y_t\right)\begin{pmatrix} I_d & 0 \\ 0 & h_\theta(Y_t) \end{pmatrix}\mathrm{d}\begin{pmatrix} W \\ W \end{pmatrix}_t \quad (7)$$

As it can be observed, this approach introduces a second collection of parameters $\phi$ that needs to be optimised in such a way that the dynamics of $X$ are statistically indistinguishable from the observed dynamics of the environment. Similarly to the setting of generative models, this statics-matching procedure can be achieved through the action of a discriminator, i.e. a carefully chosen set of statistics that seeks to match the distribution of trajectories sampled from model to the distribution of trajectories sampled from the environment. An optimally-trained model is one for which

$$\mathbb{E}_{X^\phi \sim \mathrm{model}}[F(X^\phi)] = \mathbb{E}_{X \sim \mathrm{environment}}[F(X)] \iff X^\phi = X \quad (8)$$

for a well-chosen class of statistics F (or 'witness functions' in the language of integral probability metrics Müller (1997)). As shown in Salvi et al. (2021b), if $F$ is chosen to be the signature (see Appendix for a definition), then the difference of the two expected signatures in equation (8) precisely defines the *signature maximum mean discrepancy* (MMD)

$$\mathcal{L}_S(X^\phi) := \left\| \mathbb{E}_{X^\phi \sim \mathrm{model}}[S(X^\phi)] - \mathbb{E}_{X \sim \mathrm{environment}}[S(X)] \right\|$$

where the norm is the $L^2$ norm over the ambient space of signatures, known as the tensor algebra (see Appendix). Consequently, we achieve the desired property through the signature MMD

$$\mathcal{L}_S(X^\phi) = 0 \iff X^\phi = X$$

Minimising $\mathcal{L}_S$ over $\phi$ is equivalent to training a moment matching network on pathspace. The signature MMD $\mathcal{L}_S$ can be computed efficiently by means of the signature kernel studied in Salvi et al. (2021a). Other choices of metrics are possible, for example using the Wasserstein distance on pathspace, as done in Kidger et al. (2021b).

With this choice of discriminator, we follow the following two-step optimisation procedure:

1. Firstly, we solve $\min_\phi \mathcal{L}_S(X^\phi)$ for a randomly initialised Neural RDE $\alpha^\theta$;

2. Secondly, we solve $\min_\theta J(\alpha^\theta)$, where the objective functional $J$, and its derivatives with respect to $\theta$, are computed using the minimiser $X^*$ of 1.

To demonstrate the effectiveness of this approach, we consider some "black-box dynamics" described by equation (9) from Sec. 5.1, and assume we are able to sample $(X, \alpha)$-trajectories from such black-box system. The encouraging results regarding the fitting of the parameters $\phi$ and $\theta$ in equation (7) using the two-step optimisation described in this section are presented in Figure 1.

---

[3]This setup encompasses a broad class of stochastic processes which we assume are general enough to model many real-world phenomena.

## 5 EXPERIMENTS

We present a number of numerical experiments demonstrating the capabilities of our method to compute approximate solutions of non-Markovian stochastic control problems in continuous-time. We benchmark the performance of our approach against a selection of alternative RNN-based models parameterising the feed-back control Han & Hu (2021). This choice of benchmarks is motivated mainly by the fact that this class of models, due to the connection between neural RDEs and RNNs, are the closest currently available alternatives to our method. The three alternative architectures we consider are: 1) RNN, 2) Long Short-Term Memory (LSTM), and 3) a Gated Recurrent Unit (GRU).

One key feature of the proposed model that we wish to study empirically is the time-resolution invariance discussed at the end of Sec. 3.2. Concretely, to test their robustness to changes in time-resolution, we train all models on a coarser time grid and then we evaluate them on a finer grid. Such a property is desirable both from the perspective of efficient training under computational budget constraints, and as an indication that the model is in fact learning a solution to the actual continuous-time problem. Lastly, a learned control that is heavily dependent on the grid that it was trained on may not be suitable for practical use; in such a case, this invariance property is critical.

Another performance criterion we will be using is the pathwise $L^2$ error between the state trajectories obtained using the NCDE control strategy and the ones obtained using the theoretical optimal control. The lower this error, the closer the trajectories sampled from the learnt state process to the trajectories of the theoretical state process.

All models are trained by sampling batches of trajectories from the state process under the parametric control, of the form given by equation (4), and then performing direct backpropagation using an Adam optimiser (Kingma & Ba, 2014) to minimise the Monte-Carlo estimate of the value of the reward functional $J(\alpha^\theta)$ in equation (5). For each experiment, the grid on which the system is simulated and the number of sample trajectories used to train and evaluate all models are kept the same. For a fair comparison, the hyperparameters of each model are adjusted such that the models all have an approximately equal number of trainable parameters. All experiments are implemented using version 1.11.0 of PyTorch and run on an NVIDIA Tesla K80 GPU. Additional experimental details can be found in the appendix.

### 5.1 STOCHASTIC CONTROL PROBLEM WITH DELAY

We consider first the example of a linear-quadratic problem with delay, also used by Han & Hu (2021). These problems are widely used to address real-world challenges. An example from the mathematical finance community is the control of intraday fill ratios when volatility is stochastic; see Cartea & Sánchez-Betancourt (2021). In their paper, the control affects the state dynamics linearly and the performance criterion is composed of a square running penalty on the control and a square running penalty on one of the entries of the state process. A detailed discussion of this problem and how an explicit solution may be obtained is given by Bauer & Rieder (2005).

With notation as before, the dynamics of the state process $X$ under a control $\alpha$ are given by

$$\mathrm{d}X_t = (A_1 X_t + A_2 Y_t + A_3 X_{t-\delta} + B\alpha_t)\mathrm{d}t + \sigma \mathrm{d}W_t, \quad t \in [0, T] \tag{9}$$

with $X_t = \phi$ for $t \in [-\delta, 0]$, $\delta > 0$ a delay parameter, the distributed delay satisfying

$$Y_t := \int_{-\delta}^{0} e^{\lambda\xi} h(X_{t+\xi})\mathrm{d}\xi, \quad t \in [0, T]$$

and the goal functional that we seek to minimise

$$J(\alpha) = \mathbb{E}\left[\int_0^T \left(Z_t^\top Q Z_t + \alpha_t^\top R \alpha_t\right)\mathrm{d}t + Z_T^\top G Z_T\right], \quad Z_t := (X_t + e^{\lambda\delta} A_3 Y_t), \ t \in [0, T].$$

The parameters $A_1, A_2, A_3 \in \mathbb{R}^{d \times d}$, $B \in \mathbb{R}^{d \times d_\alpha}$, $\sigma \in \mathbb{R}^{d \times d_W}$, $Q, G \in \mathbb{R}^{d \times d}$, $R \in \mathbb{R}^{d \times d}$, $\lambda, \delta$ and $T$ are all taken to be the same values as those used by Han & Hu (2021). In particular, the problem is considered in 10 dimensions in state, noise and control, $Q, R, G$ are proportional to identity matrices, the elements of $A_1, A_3, B$ and $\sigma$ are selected randomly and $A_2$ is determined by a condition guaranteeing an explicit solution. We refer to Han & Hu (2021) for further details. The constant initial condition $\phi$ is taken to be zero. The explicit value function and optimal control are obtained in terms of the solution to an associated Riccati equation, which can be solved numerically.

Table 1: **Linear-quadratic problem with delay**. Final estimate of the goal functional on the evaluation grid. Lower indicates smaller error. Analytical value: 2.231. Training resolution is given as a percentage of the evaluation resolution of 80 time steps.

|  | Training Resolution | | | |
| --- | --- | --- | --- | --- |
| Model | 100% | 50% | 25% | 12.5% |
| RNN | 2.493 | 5.162 | 8.870 | 7.600 |
| LSTM | 2.357 | 7.323 | 5.888 | 6.863 |
| GRU | 2.356 | 2.830 | 7.311 | 18.70 |
| Neural RDE (ours) | 2.358 | 2.457 | 2.509 | 2.803 |

The results for this experiment are shown in table 1. We see that, trained at full resolution, the LSTM, GRU and Neural RDE models all perform approximately as well. However, when the training grid is made coarser, the Neural RDE model remains relatively stable with only slight increases in error, dramatically outperforming the benchmark models whose performance rapidly deteriorates.

## 5.2 STOCHASTIC CONTROL PROBLEM DRIVEN BY FRACTIONAL BROWNIAN MOTION

Next, we demonstrate the application of the proposed method to a problem with non-Markovianity stemming from correlated noise increments by considering a linear-quadratic problem driven by fractional Brownian motion. The dynamics for the state $X$ are given by

$$\mathrm{d}X_t = (AX_t + C\alpha_t)\mathrm{d}t + \sigma\mathrm{d}W_t^H, \quad t \in [0,T], \qquad X_0 = 0 \tag{10}$$

where $A \in \mathbb{R}^{d \times d}$, $C \in \mathbb{R}^{d \times d_a}$, $\sigma \in \mathbb{R}^{d \times d_W}$ are parameters and $W^H$ is a $d_W$-dimensional fractional Brownian motion with components with Hurst parameters $H \in (0,1)$ (assumed the same across all $d_W$ channels). We choose the Hurst parameter $H = 0.3$, so as to highlight the applicability of the method also in the case where solution paths are rougher than Brownian motion ($H = 0.5$).

The quadratic cost functional is as follows

$$J(\alpha) = \frac{1}{2}\mathbb{E}\left[\int_0^T \left(X_s^\top Q X_s + \alpha_s^\top R \alpha_s\right)\mathrm{d}s + X_T^\top G X_T\right], \tag{11}$$

where $Q, R, G$ are symmetric and positive definite. We consider the problem specifically in two dimensions in both state and control and take $T = 1$, $\sigma = I$,

$$A = \frac{1}{10}\begin{pmatrix} 12 & 2 \\ 2 & 12 \end{pmatrix}, \quad C = \frac{1}{10}\begin{pmatrix} 15 & -3 \\ -3 & 15 \end{pmatrix}, \quad Q = R = G = \frac{1}{10}I.$$

Table 2: **Linear-quadratic problem driven by fractional Brownian motion**. Final estimate of the goal functional on the evaluation grid. Lower indicates smaller error. Training resolution is given as a percentage of the evaluation resolution of 40 time steps.

|  | Training Resolution | | | |
| --- | --- | --- | --- | --- |
| Model | 100% | 50% | 25% | 12.5% |
| RNN | 0.923 | 0.911 | 1.246 | 2.785 |
| LSTM | 0.873 | 0.961 | 1.779 | 3.422 |
| GRU | 0.891 | 0.925 | 1.236 | 2.791 |
| Neural RDE (ours) | 0.896 | 0.902 | 0.927 | 1.104 |

Table 2 shows the results for this experiment. We observe comparable performance between the LSTM, GRU and Neural RDE models at full training resolution, but with the Neural RDE significantly outperforming the other models when training resolution is decreased. At 12.5% of evaluation resolution, the models are trained on simulations using just five time steps; nevertheless, the Neural RDE appears to produce reasonable results with an error compared to the full resolution case more than one order of magnitude smaller than for the other models.

## 5.3 PORTFOLIO OPTIMISATION PROBLEM WITH COMPLETE MEMORY

We consider a portfolio optimisation problem with complete memory also studied in Han & Hu (2021). A detailed analysis of this problem including derivations of explicit solutions under exponential, power and log utilities is given in Pang & Hussain (2017). Here, the state process $X_t$ represents the wealth of an investor and the $\alpha_t = (\alpha_t^1, \alpha_t^2)$ is a 2-dimensional control process, where

$\alpha_t^1$ is the amount of investment and $\alpha_t^2$ is the consumption of the underlying asset, i.e. the fraction of wealth consumed at time $t$. The dynamics are given, for $t \in [0, T]$, by

$$dX_t = (((\mu_1 - r)\alpha_t^1 - \alpha_t^2 + r)X_t + \mu_2 Y_t)dt + \sigma\alpha_t^1 X_t dW_t, \quad Y_t := \int_{-\infty}^0 e^{\lambda\xi} X_{t+\xi} d\xi, \quad (12)$$

with $X_0 = \phi(0)$, $Y_0 = \int_{-\infty}^0 e^{\lambda\xi} \phi(t+\xi)d\xi$, for some square integrable function $\phi$. The goal functional that we seem to maximise is as follows

$$J(\alpha) = \mathbb{E}\left[\int_0^T e^{-\beta t} U_1(\alpha_t^2 X_t)dt + e^{-\beta T} U_2(X_T, Y_T)\right],$$

where $U_1(x) = \log(x), U_2(x, y) = \frac{1}{\beta}\log(x + \eta y), \eta = \frac{1}{2}(\sqrt{(r + \lambda^2) + 4\mu_2} - (r + \lambda))$. As in the previous example all the parameters are taken to be the same as in Han & Hu (2021).

Table 3: **Portfolio optimisation with complete memory**. Relative difference between the estimated and the theoretical goal functionals as well as relative pathwise $L^2$ error between the true and estimated process trajectories. Lower indicates smaller error.

|  | Relative errors ($\times 10^{-3}$) | |
| --- | --- | --- |
| Model | Goal functional | Pathwise $L^2$ |
| RNN | 0.262 | 0.555 |
| LSTM | 1.034 | 2.929 |
| GRU | 0.541 | 1.116 |
| Neural RDE (ours) | **0.238** | **0.043** |

The results for this experiment are shown in table 3, where we report the relative difference between the estimated and the theoretical goal functionals as well as relative pathwise $L^2$ error between the true and estimated process trajectories. We can see that the Neural RDE model slightly outperforms all alternative models on the relative difference of goal functionals and outperforms the second best model by one order of magnitude on the pathwise $L^2$ error.

## 6 CONCLUSION

We proposed a framework for solving non-Markovian stochastic control problems continuous-time leveraging Neural RDEs. The main idea consists in parameterising the control process as the solution of a Neural RDE driven by the state process, so that the control-state joint dynamics are governed by an uncontrolled RDE with vector fields parameterised by neural networks. To deal with input paths of infinite 1-variation, we prove Theorem 3.1 which extends the universal approximation result in Kidger et al. (2020) to Neural RDEs driven by random rough paths. We showcased the time-resolution-invariance of our approach on various non-Markovian problems, achieving better performance than traditional RNN-based approaches. Finally, we discussed possible extensions of this framework to the setting of non-Markovian continuous-time reinforcement learning and provide promising empirical evidence in this direction.

### 6.1 LIMITATIONS AND FUTURE WORK

**Path-dependent PDEs-BSDEs** As highlighted in the introduction, stochastic control problems are intimately linked with PDEs and BSDEs. The path-dependent case is still an active area of research, both from the theoretical and numerical standpoints. We find that extending methods such as Han et al. (2017) and Sirignano & Spiliopoulos (2018) to the case of path-dependent or fractionally driven control problems could be an interesting future research direction.

**Exploration-exploitation trade-off** An analysis of exploration-exploration trade-off is an important component of many RL algorithms. In our setting we opted for a simple explore-then-commit strategy, where exploration is carried out in full first during training (no exploitation), and exploitation is performed only during evaluation. Rigorously analysing the exploration-exploitation trade-off requires quantifying the finite-sample accuracy of the estimated Neural SDE, and the sensitivity of the neural CDE policy with respect to the underlying model (Szpruch et al., 2021). Both complements are technically involved in the present non-Markovian setting. Nonetheless, as mentioned in the abstract and introduction, the RL component of our paper (model-free approach) constitutes what we believe being a promising extension of the model-based framework that we hope will be leveraged and fully analysed by the RL community in the near future.

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

# A APPENDIX

## A.1 BACKGROUND ON ROUGH PATH THEORY

The purpose of this appendix is to give an informal and concise introduction to rough paths, their signatures, and their applications to machine learning.

For the abstract theory of rough paths we refer to Friz & Victoir (2010); Friz & Hairer (2020). An $\alpha$-Hölder *rough path* $\boldsymbol{X}$ consists of an $\alpha$-Hölder continuous path $X \colon [0, T] \to \mathbb{R}^d$ (the *trace* of $\boldsymbol{X}$) together with a collection of higher-order functions defined on the simplex $\Delta[0, T] \coloneqq \{(s, t) \in [0, T]^2 \mid 0 \leq s \leq t \leq T\}$ which represent, in a precise algebraic and analytic sense, iterated integrals of $X$ against itself. When $X$ is smooth or of bounded variation, such integrals can be defined canonically in the usual Stieltjes sense, and similarly when $X$ is $1/2 < \alpha$-Hölder continuous they can be defined canonically via Young integration. However, when $\alpha \leq 1/2$ there is no canonical way of defining them, and if $X$ is a stochastic process, $\boldsymbol{X}$ is often defined through some notion of stochastic integration such as Itô or Stratonovich. $\boldsymbol{X}$ takes values in $T^{\lfloor 1/\alpha \rfloor}(\mathbb{R}^d)$, where $T^N(\mathbb{R}^d) \coloneqq \bigoplus_{n=0}^N (\mathbb{R}^d)^{\otimes n}$ denotes the tensor algebra over $\mathbb{R}^d$ truncated at level $N$ and $\lfloor \cdot \rfloor$ is the floor function: this means, the rougher $X$ is, the more terms $\boldsymbol{X}$ must contain. Once such terms are defined, the *signature* $\mathcal{S}(\boldsymbol{X})$ of $\boldsymbol{X}$ is canonically defined through well-known notions of path integration. $\mathcal{S}(\boldsymbol{X})$ is a map $\Delta[0, T] \to T((\mathbb{R}^d))$ (the algebra of formal series of tensors), and when $\alpha > 1/2$ it is canonically defined by Young integration as

$$\mathcal{S}(\boldsymbol{X})_{st}^{(n)} \coloneqq \int_{s < u_1 < \ldots < u_n < t} \mathrm{d}X_{u_1} \otimes \cdots \otimes \mathrm{d}X_{u_n}$$

where the superscript $(n)$ denotes projection onto $(\mathbb{R}^d)^{\otimes n}$. When $\alpha \leq 1/2$ the whole of $\boldsymbol{X}$, not just the trace $X$, is needed to define $\mathcal{S}(\boldsymbol{X})$, and $\mathcal{S}(\boldsymbol{X})^{(n)} = \boldsymbol{X}^{(n)}$ for $n \leq \lfloor 1/\alpha \rfloor$. We will denote $\mathscr{C}^\alpha([0, T], \mathbb{R}^d)$ the metrisable topological space of $\alpha$-Hölder rough paths taking values in $\mathbb{R}^d$ with time horizon $T$: this is what Friz & Victoir (2010) call $C^{\alpha\text{-Höl}}([0, T], G^{\lfloor 1/\alpha \rfloor}(\mathbb{R}^d))$; in Friz & Hairer (2020) (which only treats the case of $\alpha > 1/3$, nevertheless sufficient for Brownian motion, which is $\alpha$-Hölder regular for any $\alpha < 1/2$) this space is denoted $\mathscr{C}_g^\alpha([0, T], \mathbb{R}^d)$, the superscript $g$ standing for "geometric". Geometric rough paths are those which satisfy integration by parts relations, and are the only ones considered here; for example, Itô and Stratonovich integration both define rough paths above Brownian motion, but only the latter is geometric. This is not an issue when considering Itô SDEs, however, which can canonically be rewritten in Stratonovich form. The main example of rough path that we will consider is the Stratonovich Brownian rough path augmented with time: if $W$ is a $d$-dimensional Brownian motion, we take $\alpha$ to be any real number in $(1/3, 1/2)$ and for $i, j = 1, \ldots d$ we let $\boldsymbol{W}_{st}^{ij} \coloneqq \int_{s < u < v < t} \circ \mathrm{d}W_u^i \circ \mathrm{d}W_v^j$, where $\circ \mathrm{d}W$ denotes Stratonovich integration. Time will take the zero-th coordinate, which means that when $i$ or $j$ above is $0$, the integral is defined through standard Young/Stieltjes integration.

The main purpose of rough path theory is to give meaning to *rough differential equations* (RDEs) $\mathrm{d}Y = V(Y)\mathrm{d}\boldsymbol{X}$ which, in addition to having usual existence and uniqueness theorems, have the property that the solution map $\boldsymbol{X} \mapsto Y$ is continuous. This is not the case when considering SDEs: the map sending the Brownian sample path to the corresponding path of the solution, though well-defined and measurable, is not continuous. An important RDE is the one satisfied by the signature itself on $T((\mathbb{R}^d))$: given a rough path $\boldsymbol{X}$ it holds that

$$\mathrm{d}\mathcal{S}(\boldsymbol{X})_{0t} = \mathcal{S}(\boldsymbol{X})_{0t} \otimes \mathrm{d}\boldsymbol{X}_t \tag{13}$$

The study of signatures is somewhat independent from that of rough paths, and is interesting even in the case of smooth or bounded variation paths (in which case $\boldsymbol{X} = X$). The main property of interest of the signature, established in Hambly & Lyons (2010) (and extended to the full rough path case in Boedihardjo et al. (2016)), is that, for paths of bounded variation, the series of tensors $\mathcal{S}(X)_{0T}$ determines the path $X$ up to *treelike equivalence*. Roughly speaking, the latter means that if two paths $X, Y$ are such that $X \star \overleftarrow{Y}$ — with $\star$ denoting path concatenation and $\overleftarrow{\phantom{Y}}$ denoting path inversion — is a path that retraces itself and returns to the starting point, then the signature will not distinguish them: $\mathcal{S}(X)_{0T} = \mathcal{S}(Y)_{0T}$. We will write $X \sim Y$ for treelike equivalence (and similarly $\boldsymbol{X} \sim \boldsymbol{Y}$ in the generalised rough path sense of Boedihardjo et al. (2016)), and note that this includes (but is not limited to) the case in which $Y$ is a reparameterisation of $X$. "Generic" paths $\mathbb{R}^d$ valued

paths can be expected not to be tree-like (i.e. not to retrace themselves) when $d > 1$; for example, in Le Jan & Qian (2013) it was shown that Brownian rough paths in dimension 2 or greater a.s. do not contain tree-like pieces.

The result of Hambly & Lyons (2010) is a powerful statement that makes it possible to understand a path $X \colon [0, T] \to \mathbb{R}^d$ in terms of the series of tensors $\mathcal{S}(X)_{0T}$. What's more, the signature has the property of "linearising" all functions on paths: any non-linear function of $X$ can be approximately expressed as a linear functional on $\mathcal{S}(X)$. A precise version of this statement in the random rough path case is proved in A.1 below. A fundamental ingredient for proving this type of result is the Stone-Weierstrass theorem: given a compact Hausdorff topological space $K$ and a subalgebra $A$ of $C(K, \mathbb{R})$ which contains a non-zero constant function and separates points (this means that for any two distinct $x, y \in K$ there exists $a \in A$ s.t. $a(x) \neq a(y)$), it holds that $A$ is dense in $C(K, \mathbb{R})$. The prototypical application of this theorem is the proof of density of polynomials in $C([a, b], \mathbb{R})$. An important property that makes it possible to apply it to signatures is that linear functions on the signature, just like polynomials, form an algebra: if $\ell_1, \ell_2 \colon T(\!(\mathbb{R}^d)\!) \to \mathbb{R}$ are linear maps then

$$\langle \ell_1, \mathcal{S}(\boldsymbol{X})_{0T} \rangle \langle \ell_1, \mathcal{S}(\boldsymbol{X})_{0T} \rangle = \langle \ell_1 \shuffle \ell_2, \mathcal{S}(\boldsymbol{X})_{0T} \rangle$$

where $\shuffle$ is the combinatorial operation of shuffling. This relation can be understood as a generalised integration by parts relation, as can be seen by taking $\ell_1$ and $\ell_2$ to be evaluations against elementary tensors: in this case (and $X$ of bounded variation) the above identity reads

$$\left( \int_{s < u_1 < \ldots < u_n < t} \mathrm{d}X_{u_1}^{i_1} \cdots \mathrm{d}X_{u_m}^{i_m} \right) \left( \int_{s < v_1 < \ldots < v_n < t} \mathrm{d}X_{v_1}^{j_1} \cdots \mathrm{d}X_{v_n}^{j_n} \right)$$

$$= \sum_{\boldsymbol{k} \in \mathrm{Sh}(\boldsymbol{i}, \boldsymbol{j})} \int_{s < r_1 < \ldots < r_{n+m} < t} \mathrm{d}X_{r_1}^{k_1} \cdots \mathrm{d}X_{r_{m+n}}^{k_{m+n}}$$

where we are summing over all multiindices $\boldsymbol{k}$ obtained by shuffling the multiindices $(i_1, \ldots, i_m)$ and $(j_1, \ldots, j_m)$. For these reasons, signatures have been extensively used for in the context of machine learning for time series, see e.g. Chevyrev & Kormilitzin (2016); Fermanian (2021b).

## A.2 Proof of 3.1

We will now prove a density result for linear functionals on the signature in a precise probabilistic sense; this is what provides motivation for considering neural SDEs as universal approximators. Let $\alpha \in (0, 1]$ and $\boldsymbol{X} \colon \Omega \times [0, T] \to T^{\lfloor 1/\alpha \rfloor}(\mathbb{R}^{1+d})$ be a stochastic $\alpha$-Hölder rough path with the property that the zero-th component of its trace is the time coordinate, $X_t^0 = t$, and whose higher components that involve the zero-th are defined canonically through Stieltjes integration. The following is the probabilistic analogue of a well-known property of the deterministic signature (see, for instance Fermanian (2021a)); it is not to be confused with the universality property of the expected signature (Lemercier et al., 2021, Theorem 3.2) with respect to functions of distributions on paths.

**Theorem A.1.** *Let $\beta < \alpha$ and $F \colon \mathscr{C}^\beta([0, T], \mathbb{R}^{1+d}) \to \mathbb{R}$ be a continuous map. Then for each $\varepsilon, \delta > 0$ there exists a truncation level $N$ and a linear map $\ell \in T^N(\mathbb{R}^{1+d})^*$ such that*

$$\mathbb{P}\big[ |F(\boldsymbol{X}) - \langle \ell, \mathcal{S}^N(\boldsymbol{X})_{0T} \rangle| \geq \varepsilon \big] < \delta \tag{14}$$

*Proof.* Let $D_r^\alpha$ be the closed disk centred at the 0 rough path, of radius $r > 0$ in $\mathscr{C}^\alpha([0, T], \mathbb{R}^{1+d})$. Since $\boldsymbol{X}$ is $\alpha$-Hölder continuous in the rough path sense

$$\lim_{r \to \infty} \mathbb{P}[\boldsymbol{X} \in D_r^\alpha] = \mathbb{P}\Big[ \bigcup_{r>0} \boldsymbol{X}^{-1}(D_r^\alpha) \Big] = \mathbb{P}\Big[ \boldsymbol{X}^{-1}(\bigcup_{r>0} D_r^\alpha) \Big] = \mathbb{P}[\boldsymbol{X} \in \mathscr{C}^\alpha([0, T], \mathbb{R}^{1+d})] = 1$$

and thus given $\delta$ as in the statement there exists $r$ s.t. $\mathbb{P}[\boldsymbol{X} \in D_r^\alpha] > 1 - \delta$. By (Friz & Victoir, 2010, Proposition 8.17 (ii)), for any $\beta < \alpha$, $D_r^\alpha$ is sequentially compact in $\mathscr{C}^\beta([0, T], \mathbb{R}^{1+d})$, and thus compact since this is a metric space. Let $\widetilde{D}_r^\alpha$ be the intersection of $D_r^\alpha$ with the aforementioned set of rough paths in $\mathscr{C}^\alpha([0, T], \mathbb{R}^{1+d})$ whose zero-th coordinate is time $t$; this is a closed set and thus $\widetilde{D}_r^\alpha$ is still compact. Thanks to the inclusion of the time coordinate, linear functions on the signature separate points in $\widetilde{D}_r^\alpha$, and by the Stone-Weierstrass theorem applied to $F|_{\widetilde{D}_r^\alpha}$ there exist $N$ and $\ell$ as in the statement s.t. $|F(\boldsymbol{X}(\omega)) - \langle \ell, \mathcal{S}^N(\boldsymbol{X}(\omega))_{0T} \rangle| < \varepsilon$ for $\omega \in \Omega$ s.t. $\boldsymbol{X}(\omega) \in D_r^\alpha$, and the conclusion now follows. $\square$

The choice for $F$ that we have in mind is the solution map of an SDE, or equivalently of an RDE driven by the enhanced Brownian rough path. The reason for the lowering of the Hölder exponent from $\alpha$ to $\beta$ lies in the cited compactness result, but does not affect the validity of applying the theorem with this choice of $F$.

*Remark* A.2 (Universal approximations of neural RDEs in probability). It follows immediately from the theorem above, the fact that the signature of a stochastic process satisfies a linear SDE, and the fact that solutions of SDEs are continuous images of functions on the driving rough path, that neural SDEs parametrised by feedforward neural nets with linear activations are dense in probability (in the same probabilistic sense of equation (14)) among all continuous functions on rough paths. In practice, one can expect superior performance (e.g. lower dimensions involved) when using non-linear activations, even though this is not needed for the theoretical result. This is because the non-linearity is already contained in operation of solving the SDE.

### A.3 Additional experimental details

In this final section of the appendix we present additional experimental details.

**Stochastic control problem with delay (sec. 5.1)** Trajectories of equation (9) are simulated using an Euler-Maruyama-type scheme on a uniform grid. All models are trained over 300 batches of 256 sample trajectories simulated on grid with 80, 40, 20 and 10 time steps. The final evaluation estimates of the goal functional are computed using 4096 sample trajectories simulated on a grid with 80 time steps. The dimension of the hidden states in the baseline models are: 400 for the RNN, 200 for the LSTM and 230 for the GRU. The latent dimension of the Neural RDE model is 200 and the vector field and initial lift are parameterised by fully connected feed-forward neural networks with two hidden layers of width $64$. We take activations given by elementwise application of the SiLU function $x \mapsto \frac{x}{1+e^{-x}}$ and apply a final `tanh` non-linearity to the outputs to prevent unreasonably large values and initial losses.

**Stochastic control problem driven by fractional Brownian motion (sec. 5.2)** Trajectories are simulated using an Euler-Maruyama scheme with increments of fractional Brownian motion sampled using the Python package fbm (Flynn). We use uniform grids with 40 time steps for evaluation and 40, 20, 10 and 5 steps for training. All models are trained over 300 batches of 256 sample trajectories. The final evaluation estimates of the value of the goal functional are computed using 4096 sample trajectories. The dimension of the hidden states in the baseline models are: 250 for the RNN, 130 for the LSTM and 150 for the GRU. The latent dimension of the Neural RDE model is 200 and the vector field and initial lift are parameterised by feed-forward neural networks with two hidden layers of width $64$ respectively.

**Portfolio optimisation problem with complete memory (sec. 5.3)** We simulate trajectories of equation (12) using an Euler-Maruyama-type scheme on a uniform grid of 200 time steps. All models are trained over 500 batches of 256 sample trajectories (similarly for evaluation). The dimension of the hidden states in the baseline models are: $600$ for the RNN, $300$ for the LSTM and $300$ for the GRU. The latent dimension of the Neural RDE model is $350$ and the vector field and initial lift are parameterised by feed-forward neural networks with two hidden layers of width $128$.

