# OpenReview forum: "Solving and Learning non-Markovian Stochastic Control problems in continuous-time with Neural RDEs"
_ICLR.cc/2023/Conference — Submitted to ICLR 2023_

### Official Review · Reviewer_8dWF · 2022-10-24

**Confidence:** 4
**Correctness:** 3
**Technical Novelty And Significance:** 3
**Empirical Novelty And Significance:** 3
**Recommendation:** 5

**Clarity, Quality, Novelty And Reproducibility:**

The paper is clear and well-written, and demonstrates an acceptable presentation, except the missing technical explanations.

**Strength And Weaknesses:**

S1- The paper considers an important problem that might be a matter of concern in different applied settings.

S2- Although there are a few typos and long sentences, the paper is well-written and the storyline is clear.

S3- The numerical experiments are discussed clearly.


W1- On the theoretical side, the paper is weak. The informal version of the main (or only) theoretical result is not informative.

W2- In the numerical experiments, only a finite memory is considered, which does not capture the general non-Markovianity, and is reducible to a Markovian problem.

W3- Several technical terms are not defined, including but not limited to the frequently used (N) RDEs.

W4- Restriction of the control policy to a parameterized family with such structures need much intuitive and formal motivations that none is provided.

W5- The incentive for following such a computationally heavy approach where much faster approximation procedures that are capable of fine tuning are not discussed, is a missing point.

**Summary Of The Paper:**

The paper considers the class of non-Markovian stochastic control problems in continuous time. In particular, two cases have been investigated. In the first case, the drift and diffusion coefficients are path-dependent. In the second, a fractional Brownian motion captures the randomness. The paper proposes a numerical approach to finding the optimal policy using neural rough differential equations (Neural RDEs). In the next step, the same problems have been considered under the situation that the system's true dynamics are not known. Although the "REINFORCEMENT LEARNING" term has been used to describe this problem, as the estimation and optimization steps are not taken simultaneously (see the last paragraph of page 6), it is important to note that there is no exploration vs exploitation trade-off present. Finally, according to the numerical examples, the proposed scheme in the paper outperforms the previous approaches.

**Summary Of The Review:**

In general, the paper provides an idea to numerically solve non-Markovian stochastic control problems in continuous time, but further required elaborations are missing. I also found the literature review a little outdated/incomplete. I will try to finalize my decision after the rebuttal period to one of the accept or reject options.

_________________________________
post rebuttal: I did not find the rebuttal convincing enough, and edited my review with further informations that are required but missing in the paper.

---

> ### Author Response · Authors · 2022-11-15
> **Response to Reviewer 8dWF**
>
> Thank you for the useful feedback and constructive criticisms. Please find below our responses to the raised concerns.
>
> **Theoretical contribution**
>
> Our main theoretical contribution extends the universal approximation result in [1] to a probabilistic density result for Neural RDEs driven by random rough paths. The interpretation is that one can approximate continuous feed-back controls arbitrarily well in probability. This extension becomes crucial in our setting because of the lack of compactness on pathspace where the Neural RDE acts on, and the infinite variation of the driving (rough) paths. These are two crucial assumptions in [1] that are violated in our setting.
>
> This result is stated informally in Theorem 3.1 in Section 3.3. and restated rigorously in the appendix. Our intention was to make the main body of the paper accessible to a broader audience and delegate the mathematical details to the appendix. If Reviewer 8dWF believes it would be better to have the rigorous statement in the main body we would be happy to move it in the camera-ready version.
>
> **Exploration-exploitation trade-off**
>
> Regarding the exploration-exploitation trade-off, we refer Reviewer 8dWF to the response given to Reviewer kQGa.
>
> **References**
>
> [1] Kidger, Patrick, et al. "Neural controlled differential equations for irregular time series." Advances in Neural Information Processing Systems 33 (2020): 6696-6707.

---

### Official Review · Reviewer_kQGa · 2022-10-24

**Confidence:** 3
**Correctness:** 3
**Technical Novelty And Significance:** 3
**Empirical Novelty And Significance:** 2
**Recommendation:** 5

**Clarity, Quality, Novelty And Reproducibility:**

- The paper is written pretty well.
- The results are stated clearly
- The setup is novel and interesting, though, the algorithms are pretty naive and not very novel.
- The algorithms should be pretty easy to implement and therefore the results should be reproducible

**Details Of Ethics Concerns:**

-

**Strength And Weaknesses:**

The setup of including the whole path of the stochastic process into the stochastic optimal control problem is new to me. This setup is probably interesting for many applications.
The theoretical contribution is solid as far as I can tell.
However, I think the developed algorithm is rather simple and pretty naive.
For example, I think that directly differentiating through the loss is usually not a good idea, as it does not scale very well and something like an adjoint method should be used, see, e.g.,  [1]. The model-based RL setup is also very crude. The proposed inference setup is similar to a method of moments. However, I think maximum likelihood estimation or a full Bayesian setup should be preferred for model estimation, see, e.g. dual control [2] or Bayesian RL [3]. It is not discussed how exploration and exploitation are balanced, though it is a very important component for model-based RL.
The presented synthetic control problems are interesting, though, I would have hoped for a nonlinear problem and a bit more motivated real-world example. Overall I think that the contribution is a bit too marginal.

Typos:
- below equation (1) the definition of $\mu$ and $\sigma$ is not correct (wrong output space for $\mu$)
- Sec 3.2 spaces are not correctly defined (wrong dimension parameters)

[1] Kidger, Patrick, et al. "Efficient and accurate gradients for neural sdes." Advances in Neural Information Processing Systems 34 (2021): 18747-18761.
[2] Stengel, Robert F. Optimal control and estimation. Courier Corporation, 1994.
[3] Ghavamzadeh, Mohammad, et al. "Bayesian reinforcement learning: A survey." Foundations and Trends® in Machine Learning 8.5-6 (2015): 359-483.

**Summary Of The Paper:**

The paper discusses a non-Markovian stochastic optimal control problem, with path-dependent dynamics and costs. It discusses the modeling of such problems and presents a numerical method based on a policy optimization procedure. For unknown dynamics, a model-based RL setup is considered, where the algorithm simultaneously learns the dynamics and optimizes the policy. The method is evaluated on two synthetic non-Markovian stochastic control problems.

**Summary Of The Review:**

The paper presents an interesting setup. The results are solid, however, in my opinion, the presented algorithms are pretty naive and are missing many modern aspects of optimal control and reinforcement learning.

---

> ### Author Response · Authors · 2022-11-15
> **Response to Reviewer kQGa (1/2)**
>
> Thank you for the useful feedback and constructive criticisms. Please find below our responses to the raised concerns.
>
> **Differentiating through the loss**
>
> Training a neural differential equation models requires backpropagating through the objective function and, by the chain rule, through the differential equation solver. There are several ways to do this. In particular, the main two approaches are:
>
> • *Discretise-then-optimise*:  this method directly backpropagate through the internal operations of the differential equation solver; this is the methodology we use in the paper. Gradient computations are accurate and fast since the underlying autodifferentiation libraries (in our case PyTorch) may better exploit parallelism. However, this approach is memory inefficient as every internal operation of the solver must be recorded, like in all classical neural network architectures.
>
> • *Optimise-then-discretise (or continuous adjoint method)*: this approach leverages a backwards in-time differential equation, solved numerically so that the forward computations need not be stored as they are recomputed on the backward pass. This approach is memory efficient, however it results in slow computations of gradients since it incurs additional computations necessary to recalculate the variables of the forward pass during the backward pass. It also produces less accurate estimates of the gradients since the latter are obtained by solving an additional differential equation that is subject to discretisation errors. In general, this means that training may be slower, final model performance may be impacted, and in the worst case, training may fail altogether. [1] gives a description of the possible failure modes of the continuous adjoint method, and [2] perform a thorough empirical investigation comparing optimise-then-discretise against discretise-then-optimise, in favour of the latter (ours).
>
> We note that in our setting, the differential equations we consider are path dependent and classical libraries (such as torchsde, diffrax etc.) cannot be directly used to perform forward and backward passes these equations. In effect, we implemented our own solver (Euler scheme). For this reason, the adjoint equations will be significantly more complex, involving functional derivatives and not available off the shelves [3].
>
> **Additional experiment**
>
> As mentioned in the response to Reviewer v9Ub, to showcase the capabilities of our model beyond the setting of non-Markovian linear quadratic problems we have added an additional experiment on portfolio optimisation in Section 5.3 of the updated version of the paper (see attached).
>
> **Alternative evaluation methods**
>
> We thank the reviewer for pointing out the references on maximum likelihood estimation and Bayesian setup. However, we do not see how these approaches could be applied to our model-based non-Markovian setting. In the model-free setting we consider, an approach like maximum likelihood cannot be used because we are considering probability measures supported on some infinite dimensional space of paths (or stochastic processes), with no meaningful definition of Lebesgue measure, and therefore no notion of likelihood. Perhaps the reviewer could be more specific on how they suggest using the mentioned techniques within our approach?
>
> **Exploration-exploitation trade-off**
>
> We agree with Reviewer kQGa that an in-depth analysis of exploration-exploration trade-off is an important component of many RL algorithms. In our setting we opted for a simple explore-then-commit strategy, where exploration is carried out in full first during training (no exploitation), and exploitation is performed only during evaluation. Rigorously analysing the exploration-exploitation trade-off requires quantifying the finite-sample accuracy of the estimated Neural SDE, and the sensitivity of the neural CDE policy with respect to the underlying model. Both complements are technically involved in the present non-Markovian setting [4]. Nonetheless, as mentioned in the abstract and introduction, the RL component of our paper (model-free approach) constitutes what we believe being a promising extension of the model-based framework that we hope will be leveraged and fully analysed by the RL community in the near future. We propose to add the above paragraph to the final section of the revised version of the paper "Limitations and future work" (see attached).

---

> > ### Author Response · Authors · 2022-11-15
> > **Response to Reviewer kQGa (2/2)**
> >
> > **Applicability to real-world problems**
> >
> > For applicability to real-world problems we refer Reviewer kQGa to the response given to Reviewer v9Ub.
> >
> > **Notation issues**
> >
> > • *Below equation (1) the definition of μ and σ is not correct (wrong output space for μ)*
> >
> > We believe the output space for μ below equation (1) is correct.
> >
> > • *Sec 3.2 spaces are not correctly defined (wrong dimension parameters)*
> >
> > We agree that the space dimensions at the beginning of section 3.2 are incorrect. We have addressed this issue in the updated version of the document (see attached).
> >
> > **References**
> >
> > [1] Gholami, Amir, Kurt Keutzer, and George Biros. "Anode: Unconditionally accurate memory-efficient gradients for neural odes." arXiv preprint arXiv:1902.10298 (2019).
> >
> > [2] Onken, Derek, and Lars Ruthotto. "Discretize-optimize vs. optimize-discretize for time-series regression and continuous normalizing flows." arXiv preprint arXiv:2005.13420 (2020).
> >
> > [3] Jazaerli, Samy, and Yuri F. Saporito. "Functional Itô calculus, path-dependence and the computation of Greeks." Stochastic Processes and their Applications 127.12 (2017): 3997-4028.
> >
> > [4] Szpruch, Lukasz, Tanut Treetanthiploet, and Yufei Zhang. "Exploration-exploitation trade-off for continuous-time episodic reinforcement learning with linear-convex models." arXiv preprint arXiv:2112.10264 (2021).

---

### Official Review · Reviewer_v9Ub · 2022-10-30

**Confidence:** 2
**Correctness:** 3
**Technical Novelty And Significance:** 3
**Empirical Novelty And Significance:** 2
**Recommendation:** 5

**Clarity, Quality, Novelty And Reproducibility:**

Stochastic differential equations and continous time control is not my field of researc so my review of this paper can unfortunately only be limited.

The paper is not written in a way that it is easy to understand. As someone who is familiar
with model-based RL, it was very hard to get an understanding of the method. For instance the introduction
of the method in 3.1 is very complicated.  Perhaps the authors could start by providing a high-level intuition of how the method
works to make it more approachable.



**Strength And Weaknesses:**

Strengths:
- The topic of the paper is in a relevant field of research: most works assume discrete time and a fully observable MDP
- The writing appears mathematically rigorous

Weaknesses:

- The paper is not written in a way that it is easy to understand. As someone who is very familiar
with model-based RL, it was very hard to get an understanding of the method. For instance the introduction
of the method in 3.1 is very complicated.
-  Experiments are restricted to artificial toy problems. While the problems may be complex I cannot access the applicability to real-world problems
- The evaluation is limited: It seems the only quality parameter the authors check for is the training resolution which would check for data efficency and robustness


Minor:
In Section 5 you write: "For a fair comparison, the hyperparameters of each model are adjusted such that the models
all have an approximately equal number of trainable parameters"

It seems you are mainly testing for data efficiency & robustness (e.g. Table 2) and on toy problems.
I don't see how using the same number of trainable parameters would be reasonable here --
because you are not testing for scalability (speed, memory footprint etc.). Perhaps some models like LSTM require more over-parameterization to work correctly.


**Summary Of The Paper:**

The authors present a method for model-based RL, particular when time is continous and the problem is partially observable. They test there method on two (toy) problems and show increased regularization and/or data efficency.

**Summary Of The Review:**

Overall, I cannot provide a strong opinion on acceptance or rejection. I cannot check the correctness of the math in Section 3. Having said that, I found the empirical part in Section 5 to be both limited (restriction to toy problems) and coarse (the only KPI being performance degradation w.r.t. training resolution).

I will assume the derivations to be correct, but because of the weaknesses in the empricial evaluation I have doubts on empirical significance.

---

> ### Author Response · Authors · 2022-11-15
> **Response to Reviewer v9Ub**
>
> Thank you for the useful feedback and constructive criticisms. Please find below our responses to the raised concerns.
>
> **Clarity of writing**
>
> In our view, Section 3.1 contains basic definitions and a minimal amount of mathematical formalism on non-Markovian stochastic control necessary to introduce our neural differential equation model in section 3.2. A “high-level intuition” of how our method works is provided in the abstract and in the paragraph “Contributions” in the introduction. To reiterate, our model parameterises the control process as a Neural RDE driven by the state process. By doing this, we can show that the control-state joint dynamics are governed by an uncontrolled RDE as per equation (4), with vector fields parameterised by neural networks. This formulation allows for trajectories sampling, Monte-Carlo estimation of the reward functional and subsequent backpropagation. We note that both Reviewers kQGa and 8dWF comment on the fact that the paper is well-written, therefore we kindly ask Reviewer v9Ub to specify which parts of Section 3.1 are unclear and we would be happy to address them in the camera-ready version of the paper.
>
> **Applicability to real-world problems**
>
> Providing accurate and fast numerical solutions to stochastic control problems is a central challenge in many areas of stochastic analysis. The type of problems we consider in this paper are of the class of non-Markovian linear quadratic problems, which are widely used to address real-world challenges ranging from mathematical finance, computational biology to robotics. We added a paragraph at the beginning of Section 5.1 of the updated versio of the paper to motivate these problems from a mathematical finance point of view (see attached).
>
> **Additional experiment**
>
> To showcase the capabilities of our model beyond the setting of non-Markovian linear quadratic problems we have added an additional experiment on portfolio optimisation in Section 5.3 of the updated version of the paper (see attached). In addition, we propose an additional performance criterion consisting in measuring the pathwise L2 error between the true and estimated process trajectories.
>
> **Evaluation**
>
> To ensure accurate approximations to solutions of control problems, computations must often be performed on very fine discretization grids, often making the overall computational cost prohibitive. Thus, it is of paramount importance in scientific machine learning (SciML) to build models exhibiting the so-called “zero-shot super resolution” property. This property consists in training a model on a course grid and then evaluating it on a finer grid without sacrificing performance. The experiments carried out in the paper show that our model is indeed resolution-invariance which means the model is more scalable than the RNNs counterparts in that it can be deployed on courser grids with an evaluation speed-up proportional to the size of the down-sampling.
>
> Regarding the memory footprint, we refer Reviewer v9Ub to the response **Differentiating through the loss** given to Reviewer kQGa.
>
> **Choice of hyperparameters**
>
> Many standard neural networks may be interpreted as approximations to neural differential equations [1]. Neural RDEs are the continuous-time limit of RNNs/LSTMs/GRUs [2]. Indeed, the two most popular types of recurrent neural networks - GRUs and LSTMs - are obtained as discretization of RDE-type models. For this reason, we believe that setting a similar number of parameters for both classes of models is a sensible choice. In our experiments, we tried to increase the parameters of RNNs models without any noticeable difference.
>
> **References**
>
> [1] Kidger, Patrick. "On neural differential equations." arXiv preprint arXiv:2202.02435 (2022).
>
> [2] Morrill, James, et al. "Neural rough differential equations for long time series." International Conference on Machine Learning. PMLR, 2021.

---

### Decision · Program_Chairs · 2023-01-20

**Decision:**

Reject

**Justification For Why Not Higher Score:**


The method is straightforward, and the design choice (e.g., policy parametrization and gradient-based planning) is unjustified.

The empirical comparison is weak on synthetic problem and lacking of competitors.

**Justification For Why Not Lower Score:**

N/A

**Metareview: Summary, Strengths And Weaknesses:**


In this paper, the authors proposed neural rough differential equations model as a model-based reinforcement learning for continuous-time and non-Markovian stochastic control problem. Particularly, the model parameters are learned by minimizing MMD, and the policy is parametrized as a recurrent function. The optimal policy is then obtained by first-order method.

There are several major issues raised by reviewers:

1, the method is relatively straightforward and the derivation is lacking of justification. Specifically,

- the justification of the parametrization of policy has not been provided;
- the gradient-based optimal policy seeking may lead to suboptimal policy due to the highly nonlinearity in the objective and dynamcis;
- there is no exploration vs. exploitation mechanism in the proposed method.

2, the empirical experiment is weak. The method is only evaluated in synthetic problem, and no standard benchmark, e.g., MuJoCo, has been tested. There is no state-of-the-art RL algorithms, e.g., [1], have been compared with. It is hard to evaluate the significance of the proposed method.

[1] Wang, Tingwu, Xuchan Bao, Ignasi Clavera, Jerrick Hoang, Yeming Wen, Eric Langlois, Shunshi Zhang, Guodong Zhang, Pieter Abbeel, and Jimmy Ba. "Benchmarking model-based reinforcement learning." arXiv preprint arXiv:1907.02057 (2019).